# Radiative impact of the Hunga stratospheric volcanic plume: role of aerosols and water vapor in the southern tropical Indian Ocean

M. Sicard[1,2], A. Baron[3], M. Ranaivombola[1], D. Gantois[1], T. Millet[1], P. Sellitto[4], N. Bègue[1], H. Bencherif[1], G. Payen[6], N. Marquestaut[6], and V. Duflot[1]

[1]Laboratoire de l'Atmosphère et des Cyclones (LACy), UMR 8105 CNRS, Université de La Réunion, Météo-France, Saint-Denis de la Réunion, 97400, France
[2]CommSensLab-UPC, Universitat Politècnica de Catalunya, Barcelona, 08034, Spain
[3]Cooperative Institute for Research in Environmental Sciences, and NOAA Chemical Sciences Laboratory, Boulder, 80305-3337, USA
[4]Université Paris Est Créteil and Université de Paris, CNRS, Laboratoire Interuniversitaire des Systèmes Atmosphériques, Institut Pierre Simon Laplace, Créteil, France
[5]Istituto Nazionale di Geofisica e Vulcanologia, Osservatorio Etneo, Catania, Italy
[6]Observatoire des Sciences de l'Univers – Réunion (OSU-R), Saint Denis, 97400, France

*Correspondence to*: Michaël Sicard (michael.sicard@univ-reunion.fr)

**Abstract.** This study attempts to quantify the radiative impact over Reunion Island (21º S, 55° E) in the southern tropical Indian Ocean of the aerosols and water vapor injected in the stratosphere by the eruption on 15 January 2022 of the Hunga underwater volcano in the South Pacific. Ground-based lidar and satellite passive instruments are used to parametrize a state-of-the-art radiative transfer model for the first thirteen months after the volcano eruption. The descending rate of the aerosol volcanic plume is -0.008 km day$^{-1}$. At this rate, aerosols are expected to be present in the stratosphere until the first half of 2025. The overall aerosol and water vapor impact on the Earth's radiation budget for the whole period is negative (cooling, -1.14 ± 0.46 W m$^{-2}$) and dominated by the aerosol impact (~96 %; the remaining ~4 % are due to the water vapor). At the Earth's surface, aerosols are the main driver and produce a negative (cooling, -1.05 ± 0.36 W m$^{-2}$) radiative impact. Between the short- (month 2 to 4 after the eruption) and mid-term (month 5 to 14 after the eruption) periods, the aerosol and water vapor radiative effect at the surface and TOA reduces 21 and 28 %, respectively. During the short-term period, a slight loss of energy of -0.18 ± 0.10 W m$^{-2}$ is observed in the stratosphere with a balanced contribution between the aerosols (60 %) and the water vapor (40 %). During the mid-term period, this effect reduces to values in the same order of magnitude than the estimated uncertainty. Heating/cooling rate profiles during the mid-term period show a clear vertical difference locally in the stratosphere between the aerosol warming impact (17 to 25 km) and the water vapor cooling one (25 to 40 km).

## 1 Introduction

More than one year and a half after the eruption of Hunga underwater volcano in the South Pacific, the scientific community is still actively investigating the climate impact of the huge amounts of water, steam and gases that were injected in the atmosphere. The event showed an extremely fast spatio-temporal, global dispersion of the stratospheric volcanic matter that

circulated the Earth in only one week (Khaykin et al., 2022) and dispersed pole-to-pole in three months (Taha et al., 2022), first in the form of concentrated patches (Legras et al., 2022). Several features are evidences of a record-breaking

atmospheric event. The eruption, equivalent to an energy of 110 Tg of trinitrotoluene (TNT) explosive, is the most powerful volcanic explosion since Krakatau (1883) and Tambora (1815) (Lac et al., 2022). The volcanic plume reached an altitude of 57 km, a coincident estimation resulting from different techniques (Carr et al., 2022; Proud et al., 2022), placing it in the upper stratosphere – lower mesosphere, a record in the satellite era. The mass of water retained in the atmosphere was unprecedented: (Millán et al., 2022) estimated to 146 Tg the mass of water injected in the atmosphere (.e.g. the 1991

Pinatubo eruption released 37 Tg of water into the atmosphere (Pitari and Mancini, 2002)). In contrast, sulfur dioxide ($SO_2$) mass injection was not that exceptional: $\sim 0.6 – 0.7$ Tg (Carn et al., 2022) which is much smaller than that from previous major eruptions (e.g. 20 Tg for Pinatubo (Bluth et al., 1992)). Still, the stratospheric aerosol optical depth (sAOD) has been recorded globally as the largest since Pinatubo eruption (Taha et al., 2022) and peaked locally at values never observed before, e.g. in the Indian Ocean (Baron et al., 2023).

The latter three variables (water, $SO_2$ and injection height), and co-emitted halogens in a lesser extent, are some of the main factors responsible for the production of volcanic sulfate and for the loss/production of ozone. The initial $SO_2$ was fully converted into sulfates in less than two weeks under the influence of water vapor (Asher et al., 2022; Legras et al., 2022), whereas volcanic sulfate and water still persist as of today. The fast water vapor injection provided abundant hydroxide (OH) which reacted with $SO_2$ to form volcanic sulfate at a faster rate than the typical $\sim 30$ days (Carn et al., 2016). Higher

concentrations of volcanic sulfate led to more rapid coagulation and thus larger particles. In the case of Hunga volcano, this mechanism is estimated to have halved the $SO_2$ lifetime and doubled the sAOD (Zhu et al., 2022). This rapid growth and global persistence of volcanic sulfate aerosols have been demonstrated with AERONET measurements by Boichu et al. (2023) with the occurrence of an unusual "volcanic fine mode" with a peak ranging in $0.28 – 0.50$ µm. This fine mode was found to be poorly absorbing, although Kloss et al. (2022) reports from balloon-borne measurements a moderately absorbing

fine mode in the first 10 days after the eruption indicating small sulfate coated ash particles. Volcanic sulfate is known to be a factor to impact ozone depletion by providing additional surface area and suppressing the nitric oxide cycle (Tie and Brasseur, 1995). The transport of volcanic sulfate from the tropics to the Antarctic by the Brewer-Dobson circulation contributed to increase ozone concentrations in the middle stratosphere but to decline in the lower stratosphere at mid-to-low latitudes (Lu et al., 2023), while, combined with a cold polar vortex, it contributed to decrease ozone concentration in the

Antarctic (Wang et al., 2022). Because ozone is not emitted primarily during volcanic eruptions, its loss or production by post-eruption reactions are more tedious to estimate (Evan et al., 2023). The effect of Hunga volcano on stratospheric ozone is still under study.

Water, volcanic sulfate and the injection height are the main drivers of the impact of Hunga volcano on atmospheric global circulation (Coy et al., 2022) and climate (Zuo et al., 2022). In particular, the climate forcing will depend on the radiative

effect produced by the water vapor longwave emission and the aerosol shortwave and longwave scattering and absorbing properties (Robock, 2000). These interaction mechanisms (emission, scattering and absorption) with the shortwave and

longwave radiation are highly height-dependent and determine the sign of the differential of energy gained (positive) or lost (negative) in all layers of the atmosphere. Several studies have demonstrated the stratospheric cooling produced by the excess of water vapor injected by Hunga volcano either locally (Sellitto et al., 2022), zonally (Schoeberl et al., 2022; Vömel et al., 2022; Zhu et al., 2022) or globally (Millán et al., 2022) at different time scales spanning from instantaneous estimates to 6-month evolutions. As far as volcanic sulfates are concerned, these aerosols usually scatter sunlight back to space, cooling the Earth's surface, and absorb outgoing thermal radiation. Several authors have made the hypothesis that Hunga volcano eruption could impact climate not through surface cooling due to sulfate aerosols, but rather through surface warming due to the radiative forcing from the excess stratospheric water vapor. The impact on the Earth's radiation budget, i.e. at the top of the atmosphere, is even more uncertain since smaller impacts, hence a greater sensitivity to variations, are at play. To date, assessments of the radiative effect of combined water vapor and aerosols have only been performed for 3 case studies during the first 10 days after the eruption by Sellitto et al. (2022), for the first two months after the eruption by Zhu et al. (2022) and for the first year after the eruption (Gupta et al., 2023). Jenkins et al. (2023) evaluated the chances of temporary global surface temperature anomaly above 1.5 ºC over the coming decade caused by Hunga volcano stratospheric water vapor perturbation.

Here, the impact of water vapor and aerosols on the Earth's radiation budget is estimated over Reunion Island (21°S, 55°E) for the first thirteen months after Hunga volcano eruption. Both water vapor and aerosols obtained from ground-based lidar and satellite measurements are used as input in a state-of-the-art radiative transfer model. The radiative effect is calculated for three scenarios considering aerosols only, water vapor only and combined aerosols and water vapor.

## 2 Materials and Methods

### 2.1 The Maïdo instrumentation

The lidar system used in this study, the Li1200 lidar (Dionisi et al., 2015; Vérèmes et al., 2019; Gantois et al., 2024), is located at the Observatoire de Physique de l'Atmosphère à La Réunion (OPAR) Maïdo station (21.079º S, 55.383º E, 2160m asl; Baray et al., 2013). The system is operating at 355 nm and measurements are made twice a week on Monday and Tuesday nights. In this work, 87 nights of observations were recorded between 19 January 2022 and 15 February 2023. A full description of the system is available in the data paper of Gantois et al. (2024).

The extinction coefficient presented in this work is obtained applying the elastic, 2-component inversion algorithm (Klett, 1985) using a constant lidar ratio (LR). Several LR values were tested between 30 and 70 sr. The value of 30 sr was fixed for this study. The transmittance method initially used in Baron et al. (2023) for the thick plume observed during the first days after the eruption over Reunion Island was not retained for at least two reasons: with decreasing aerosol loads, the transmittance method would have led to large uncertainties in the LR retrieval; the unreliability of the method for ground-based systems in low aerosol loads and at such altitude levels (17 – 32 km). Although rather unusual for sulfate aerosols which are more often associated to LR of 60 sr according to the existing literature (e.g. Lopes et al., 2019), the value of 30 sr

is chosen following the results presented in Baron et al. (2023). Indeed the latter found values of LR at 355 nm in the range 29 – 35 sr with small standard deviations (< 7 sr) by applying the transmittance method during several nights in January 2022. It reflects the uncommon properties of these volcanic particles which were proved to be stable over time by Duchamp et al. (2023) using SAGE-III (Stratospheric Aerosol and Gas Experiment) observations. The use of 30 sr at 355 nm is also coherent with Mie calculations conducted by Baron et al. (2023) and with the size distribution parameters from Duchamp et al. (2023).

The uncertainty associated to the extinction profiles at 355 nm, and by extension to the sAOD at 355 nm, has been calculated considering an uncertainty on the lidar ratio of $\pm 10$ sr. This value of $\pm 10$ sr corresponds to the largest uncertainty calculated on the lidar ratio at 355 nm by Baron et al. (2023) for the Hunga plume over Reunion Island in January 2022. To compare the lidar-derived and OMPS sAOD, the wavelength of 745 nm was used (see Section 2.2). sAOD at 355 nm was converted into sAOD at 745 nm using a constant Ångström power law, $AE_{355/745}$, of -0.14 resulting from our Mie calculation (see Section 2.3). The uncertainty associated to the sAOD at 745 nm has been calculated considering both the uncertainty associated to sAOD at 355 nm and the uncertainty associated to $AE_{355/745}$, fixed to a constant value of $\pm 1.0$ (Baron et al., 2023).

It is our belief that the results in Reunion Island can be easily generalized throughout the southern tropical Indian Ocean region. Mallet et al. (2018) reported for the first time the pristine characteristics of the southern Indian Ocean region located between 10 and 40°S and between 50 and 110°E. Except its very northern boundary, this domain is not impacted by the longitudinal transport of the Asian monsoon over the northern Indian Ocean. Tropospheric sea salt aerosols are the dominant and the AOD-modulating aerosol type (Mallet et al., 2018). The same statement is true over Reunion Island (Duflot et al., 2022). The synoptic circulation in the southern tropical Indian Ocean is strongly connected with the Mascarene anticyclone, which, because of its location in the middle of this basin, limits the transport of terrestrial aerosols to this region. Several indicators of the homogenous dispersion of the volcanic plume in the stratosphere at our latitudes are exposed in Section 3 and reinforce the assumption made from now on that the results in Reunion Island can be generalized to the whole southern tropical Indian Ocean region.

## 2.2 Satellite and reanalysis data

The Ozone Mapper and Profiler Suite Limb profiler has been on-board the Suomi National Polar Partnership (NPP) since October 2011. Using limb scattering solar radiation, OMPS provides good quality of aerosols extinction retrievals at several wavelengths: 510, 600, 675, 745, 869 and 997 nm (Taha et al., 2021). As recommended by the latter, we use data product version 2.0 of aerosol extinction profile at 745 nm to follow the aerosol volcanic plume over Reunion Island, from January 2022 to mid-April 2023. These data are provided from 10 to 40 km height on a vertical grid of 1 km. Stratospheric aerosol optical depth calculations are made by integrating the extinction profiles from 17 km to 40 km, where 17 km corresponds to tropopause height over Reunion Island (Bègue et al., 2010). Based on previous studies in the Southern Hemisphere (Bègue et al., 2017; Tidiga et al., 2022), background periods extend from 2012 to February 2014 and from January 2017 to April 2018,

to exclude volcanic eruptions (Kelud, Calbuco, Ambae and Ulawun) and Australian 2019/2020 biomass burning episode (the Black Summer).

MLS (Microwave Limb Sounder) version 5.0, level 3 data are also used to extract the monthly mean water vapor over our site and in the stratosphere during 2021 to serve as a climatological reference. See https://mls.jpl.nasa.gov/data/v5-0_data_quality_document.pdf for more details about this MLS product. The monthly mean of the water vapor in the altitude range of interest in 2021 is 4.5 ppmv. This value sets the climatological reference necessary to parametrize the unperturbed conditions of the water vapor.

The MERRA-2 Stratospheric Composition Reanalysis of Aura MLS (M2-SCREAM) products are used for characterizing the water vapor (WV) and ozone vertical distribution, in particular the 3D, 3-hourly GMAO_M2SCREAM_INST3_CHEM product available until 1 April 2023 (doi: 10.5067/7PR3XRD6Q3NQ). This product, produced at NASA's Global Modeling and Assimilation Office (GMAO), is generated by assimilating MLS and OMI (Ozone Monitoring Instrument) retrievals into the GEOS (Goddard Earth Observing System) Constituent Data Assimilation System (CoDAS) driven by meteorological fields from MERRA-2. Stratospheric water vapor and ozone, among other compounds, are assimilated in M2-SCREAM. Assimilated fields are provided globally at 0.5° (latitude) by 0.625° (longitude) resolution from approximately 10 km up to the lower thermosphere. Concretely, the variables of specific humidity (QV, kg kg$^{-1}$), ozone (O3, ppmv), mid-layer pressure (PL, Pa) and mid-layer height (H, m) were used. The specific humidity was converted to the actual water vapor pressure and then to water vapor mixing ratio. All variables where averaged over four pixels surrounding the Maïdo coordinates. Assimilation uncertainties for each of the assimilated constituents are calculated from the CoDAS statistical output (Wargan et al., 2023). For the period January 2022 to September 2022 and in the height interval of interest of this study (17 – 32 km) the uncertainty on the water vapor and ozone are less than 0.2 and 0.13 ppmv (Wargan et al., 2023), respectively.

### 2.3 The GAME radiative transfer model

#### 2.3.1 Code and parametrization

Radiative fluxes propagating through the atmosphere were calculated with the radiative transfer (RT) model GAME (Dubuisson et al., 1996; Dubuisson, 2004; Dubuisson et al., 2006). For this study, GAME was set up to calculate spectrally integrated upward and downward radiative fluxes in 40 plane and homogeneous layers from 0 to 100 km with a 1 km resolution from 0 to 30 km and a coarser resolution above. The shortwave (SW) spectral range was set from 0.2 to 4.0 μm (wave number resolution of 400 cm$^{-1}$ from 0.2 to 0.7 μm and 100 cm$^{-1}$ from 0.7 to 4.0 μm). In the longwave (LW) spectral range, spectral limits were defined between 4.0 and 50.0 μm (115 points at a wave number resolution of 20 cm$^{-1}$). GAME calculates solar flux values at the boundary of plane and homogenous atmospheric layers by using the discrete ordinates method (Stamnes et al., 1988). Gas ($H_2O$, $CO_2$, $O_3$, $N_2O$, CO, $CH_4$, and $N_2$ are considered) absorption is calculated from the correlated $k$ distribution (Lacis and Oinas, 1991). More details about the computation of the gas transmission functions can

be found in (Dubuisson, 2004) and (Sicard et al., 2014). In the longwave spectral range, GAME presents the advantage of the complete representation of the long-wave aerosol scattering, in addition to their absorption (Sicard et al., 2014).

For the sake of clarity and comparability with other works, we recall the definition of the direct radiative effect ($DRE$) of a perturbed vs. unperturbed atmospheric compound on the Earth's radiation budget. At a given height level, $L$:

$$DRE(L) = \left[F_p^\downarrow(L) - F_p^\uparrow(L)\right] - \left[F_u^\downarrow(L) - F_u^\uparrow(L)\right] \qquad (1)$$

where $F$ are the radiative flux values for the perturbed ($p$ subindex) and unperturbed ($u$ subindex), while the $\downarrow$ and $\uparrow$ arrows indicate, respectively, the downward and upward flux direction. By that definition, negative (positive) $DRE$ values represent

a cooling (warming) effect. The $DRE$ was calculated at two climate-relevant altitude levels: at the top of atmosphere (TOA) and at the bottom of atmosphere (BOA). The contribution in the atmospheric column is quantified by the atmospheric direct radiative effect, $DRE(ATM)$, which is defined as follows:

$$DRE(ATM) = DRE(TOA) - DRE(BOA) \qquad (2)$$

As far as GAME parametrization is concerned, temperature and pressure profiles used in both SW and LW simulations are

175 taken from radiosoundings launched from Saint-Denis, the state capital of Reunion Island, 20 km North of Maïdo, every night at 00:00 Local Time. Aerosols are fully parameterized in GAME by the user in terms of spectrally and layer-mean aerosol optical depth (AOD), single scattering albedo (SSA), and asymmetry factor (asyF). The layer-mean AOD is distributed vertically according to the profiles of the measured extinction coefficient at 355 nm, whereas SSA and asyF are assumed vertically constant.

The spectral AOD, SSA and asyF were calculated in the whole spectral range with a Mie code. A monomodal, lognormal size distribution was considered with a geometric median radius of 0.35 µm and a mode width of 1.23. These values, taken from Duchamp et al. (2023), are the retrieved particle size distribution of SAGE-III in the 30º S - 10º S latitude range and corresponding to plume conditions at altitude of maximum extinction averaged between the months of June and August 2022. SAGE-III observations also show that this size distribution at the plume peak height persisted over 17 months with

only a small decreasing trend in the size. For the refractive index we used the GEISA (Gestion et Étude des Informations Spectroscopiques Atmosphériques: Management and Study of Spectroscopic Information) spectroscopic database (Jacquinet-Husson et al., 2008). In particular the refractive index of the binary system $H_2SO_4/H_2O$ with a $H_2SO_4$ mixing ratios (in mass, $w_t$, i.e., the ratio of the $H_2SO_4$ mass to the total mass of the droplets) of 0.75 and at temperature of 215 K (this temperature corresponds in average to the atmospheric temperature at the height of the volcanic plume) was used. This

value has been selected in view of the results from Duchamp et al. (2023) who, in their supporting information, show $w_t$ profiles retrieved from zonally average profiles of water vapour (retrieved from MLS, version 5) and temperature (from the ERA5 reanalysis) at latitudes 0, 10, 20 and 30°S for a set of dates between February 2022 and April 2023. At our latitude (20°S) $w_t$ ranges between 0.70 and 0.80 between 22 and 28 km (where the moist layer is located at least until November 2022) for all dates shown. Above 28 km, $w_t > 0.80$. The real part is defined over the range $0.61 - 5000.00$ µm (wave number

resolution of 2 cm$^{-1}$). The value at 0.61 µm was assumed constant in the range $0.20 - 0.61$ µm. The imaginary part is defined

over the range 2.36 – 23.15 μm (wave number resolution of 0.96 cm$^{-1}$). The value at 2.36 μm was assumed constant in the range 0.20 – 2.36 μm. The reader is referred to Biermann et al. (2000) for more details on this dataset. Figure 1 shows the real part (RRI) and imaginary part (IRI) of the refractive index used for $w_t$ = 0.75. The curves for $w_t$ = 0.80 are shown only for comparison. Large spectral variations in the infrared atmospheric window (8–13 μm), which have an important impact on the infrared radiative budget of the atmosphere, are visible. The most astonishing feature of the figure is probably the high absolute values of the IRI which emphasizes the high absorbing properties of sulfate aerosols in the longwave spectral range which induce extremely low SSA (< 0.2) in the whole range. One sees that between $w_t$ = 0.75 and $w_t$ = 0.80 nearly no difference is visible, so that employing $w_t$ = 0.75 at all altitudes (i.e., even above 28 km) should not impact the LW radiative properties. For comparison IRI (this study) is 2 to 3 times larger in the atmospheric window than IRI for mineral dust (Sicard et al., 2014).

The spectral SW surface albedo was interpolated from the four-wavelength AERONET L2.0 annual mean for year 2022 and assumed constant above 1020 nm. The AERONET site used is "Maido_OPAR". It is situated at 2160 m asl and is collocated to the lidar. For the LW broadband surface albedo, we used IASI (Infrared Atmospheric Sounding Interferometer) December nighttime monthly mean climatology of the surface emissivity (i.e. 1 - surface albedo) at 890 cm$^{-1}$, i.e. 11.24 μm, over the Indian Ocean (Zhou et al., 2013), and set the surface albedo value to 0.01.

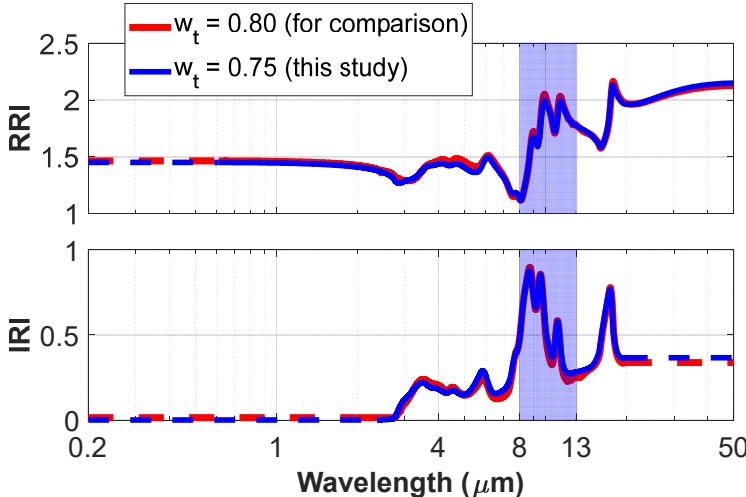

**Figure 1: Spectral complex refractive index considered for the calculation of the aerosol radiative properties in the shortwave (0.2 – 4 μm) and longwave (4 – 50 μm) spectral ranges. See text for details. The infrared atmospheric window (8–13 μm) is indicated by the blue shaded area. The dash lines are the extrapolation of the dataset used.**

So as to avoid the dependency on solar zenith angle, only daily radiative effects are presented in this work. To do so, each nighttime measurement and parametrization is assumed to be constant for the 24 hours of the day considered and both SW and LW radiative effects are calculated at an hourly time resolution between 00:00 and 23:00 UT. In these calculations, the solar zenith angle is the only parameter that varies. The daily radiative effect is the average of the 24 hourly *DRE*.

     **2.3.2 Error budget**

An error budget is performed to quantify the uncertainties made on the radiative effect estimations using GAME and caused by the model itself, our parametrization and the hypothesis made. GAME model participated to an intercomparison exercise (Halthore et al., 2005) which concluded that it is accurate to a few units of watt (< 5 W) for a flux reaching 1000 W m$^{-2}$. The impact of this uncertainty on our estimations should be even less since only daily averaged fluxes are considered. It is thus

reasonable to consider an uncertainty in relative terms of 0.5 %.

Two other sources of error are considered: one associated to the lidar ratio selected and another associated to the size distribution selected. The constant lidar ratio used in the elastic, 2-component inversion algorithm is 30 sr. Baron et al. (2023) estimated an uncertainty of ± 10 sr for the Hunga plume over Reunion Island in January 2022 (see Section 2.1). New profiles of the extinction inverted using (30 + 10) sr and (30 - 10) sr were used in GAME to quantify the deviation from the

nominal (LR = 30 sr) radiative effect estimations. As far as the size distribution is concerned, Duchamp et al. (2023) detected "a small decreasing trend in the size" without quantifying it. We have assumed a decrease of the geometric median radius of -0.01 μm. Thus, a new Mie calculation was performed with a geometric median radius of 0.34 μm and the resulting radiative properties were used in GAME to quantify the deviation from the nominal (geometric median radius of 0.35 μm) radiative effect estimations. The results from these uncertainties are given in Table 1 in relative terms at BOA and TOA and in

absolute terms in the atmosphere. Logically, the lidar ratio error which impacts almost proportionally the sAOD error is by far the strongest. We can reasonably consider that the aerosol daily radiative effects are estimated with an uncertainty better than 48 % at TOA and better than 42 % at BOA. The resulting atmospheric radiative effect (TOA – BOA, see Eq. 2) is given with an uncertainty of +0.09 / -0.06 W m$^{-2}$.

| Source of error | TOA | BOA | ATM |
|---|---|---|---|
| GAME model | < + 0.5 % | < + 0.5 % | < + 0.5 % |
| LR (+10 / -10 sr) | +47 / -40 % | +42 / -38 % | +0.09 / -0.06 W m$^{-2}$ |
| Geometric median radius (-0.01 μm) | +4 % | ~0 % | < 0.01 W m$^{-2}$ |
| Total | +48 / -40 % | +42 / -38 % | +0.09 / -0.06 W m$^{-2}$ |

**Table 1. Error budget of the aerosol daily radiative effect.**

## 3 Vertical/temporal evolution of the Hunga volcanic plume over Reunion Island

The historical context of the aerosol load over Reunion Island is shown in Figure 2 by the temporal evolution of the stratospheric AOD at 745 nm measured by OMPS in the last decade. The background sAOD is measured over the unperturbed years 2012 and 2013. It is $(2.6 \pm 0.1) \times 10^{-3}$. At each exceptional event the sAOD takes off from this background sAOD and since the eruption of Ambae in July 2018 the sAOD over Reunion Island has never turned back to its background value. The sAOD peak produced by Hunga volcano (0.035) is the highest in the last decade and it is a factor 4 times higher than the second highest event (0.009, Calbuco eruption in April 2015). Zonal averages between 30º S and 15º N for Hunga volcano and 20º S and 90º S for Calbuco showed that Hunga sAOD was more than double that for the 2015 Calbuco eruption (Taha et al., 2022). The reason why the local and zonal sAOD differences between Hunga volcano and Calbuco differ lies in the zonal mean stratospheric conditions. In the case of Hunga volcano, a marked easterly band (Khaykin et al., 2022; Legras et al., 2022) favored a direct transport from Hunga volcano towards Reunion Island (both being approximately at the same latitude). Further back historically, the 40+ year satellite record of monthly sAOD for the 60º S – 60º N latitude band in Khaykin et al. (2022) shows that only the eruptions of Pinatubo (1991) and El Chichón (1982) exceeded the Hunga one in terms of absolute stratospheric AOD (by a factor of 6 and 3, respectively).

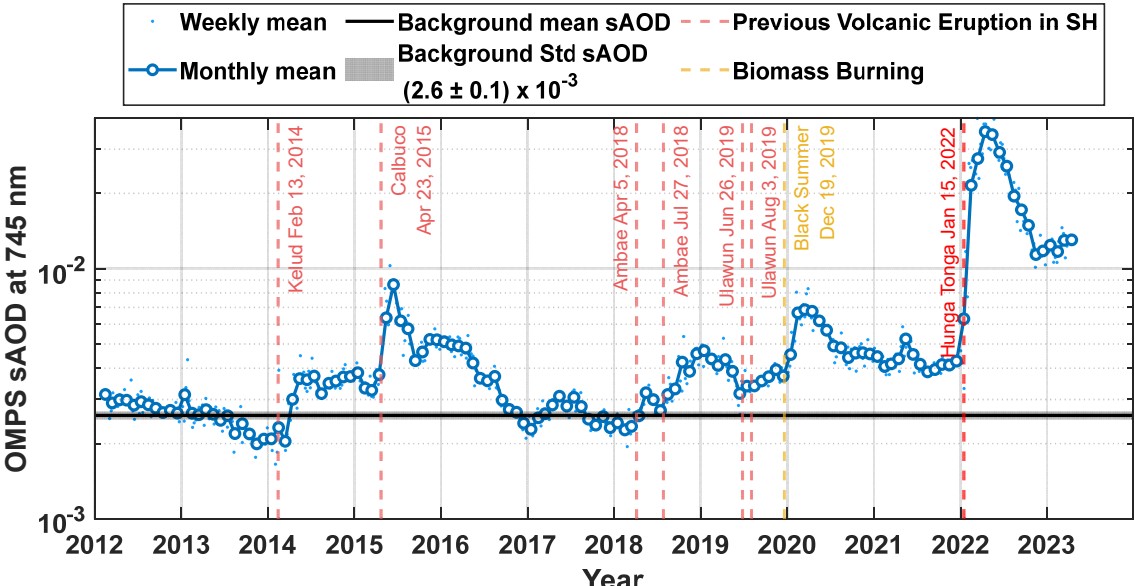

Figure 2: sAOD (17 – 40 km) at 745 nm from OMPS over Reunion Island. The most important volcanic eruptions (name and date) in the southern hemisphere are indicated by red vertical lines. The Australian 2019/2020 biomass burning episode is indicated in orange.

In Figure 3, the vertical and temporal (January 2022 – April 2023) evolution of the Hunga volcanic plume over Reunion Island is analyzed by means of sAOD at 745 nm and profiles of extinction coefficient at 745 nm, water vapor and ozone.

While the monthly OMPS sAOD peak is reached in April and May 2022, the instantaneous lidar sAOD peaks just a few days after the eruption, reaching 0.40 on 21 January. This time difference is an indication of the dispersion time of the volcanic matter injected by Hunga volcano in the stratosphere at the global scale. Other studies confirm that the volcanic plume dispersed nearly pole-to-pole in three months (Khaykin et al., 2022; Taha et al., 2022). Another indicator of this dispersion is the standard deviation (calculated as a 15-day rolling standard deviation) associated to OMPS monthly sAOD: once passed the first month, it steadily decreases all along year 2022. The agreement between monthly and instantaneous sAOD which becomes excellent as of April 2022 is also an indicator of the homogenous dispersion of the volcanic plume in the stratosphere at our latitudes. It also reinforces our belief that these results in Reunion Island could probably be generalized throughout the southern tropical Indian Ocean region. A decrease of the monthly sAOD is observed after April/May and until November. Then sAOD stabilizes until today (sAOD = 0.012, almost 5 times the background sAOD). From September 2022 on, the lidar sAOD is slightly higher than OMPS sAOD, although the error bars always overlap one another. It is not clear whether this is reflecting a systematic difference between both estimations or a limitation of one of the two datasets.

The time-height plot of the extinction coefficient (Figure 3b) shows clearly the height and vertical extension of the volcanic plume which is still present on 15 April 2023 and located at 18.5 – 23.5 km height (sAOD = 0.012). The plume peak height has a decreasing tendency since April 2022 at an average steady rate of -244 m per month or $\sim$ -0.008 km day$^{-1}$. Assuming this rate constant in time and a tropopause height in Reunion Island of 17 km (Bègue et al., 2010), the remaining life time of the volcanic plume in the stratosphere is estimated to be between 2 and 2.5 years after 15 April 2023. Except during the first week of detection above Reunion Island, the Hunga volcanic plume is not detected above 30 km. The water vapor plume (Figure 3c) reveals also clearly the unusually high water vapor concentration caused by the volcanic plume. A local peak of 65 ppmv is reached on 13 February 2022. It is almost 15 times higher than the climatological reference value of 4.5 ppmv (see Section 2.2). On a monthly basis, the water vapor stratospheric peak in February 2022 is approximately 5 times higher than the climatological reference (4.5 ppmv). This ratio decreases to almost 2 in February 2023. The water vapor plume is thinner than the aerosol one and located at a higher altitude, 3 to 4 km higher. Such a difference, although not so accentuated, is observed zonally at 15º S during the first six months of year 2022 (Schoeberl et al., 2022). The height of the peak of the aerosol and water vapor layers (respectively, red and black lines in Figure 3b and Figure 3c) have opposite tendencies as of April 2022: the aerosol plume is slowly descending whereas the moist layer is ascending slowly until October 2022 and at a higher rate afterwards. Schoeberl et al. (2022) explain that "the water vapor is transported upward with the diabatic circulation that gives rise to the tropical trace gas tape recorders (Schoeberl et al., 2018) whereas the aerosols are gravitationally settling". Legras et al. (2022) who analyze the same period (first six months of year 2022) precise that the ascent of the moist layer is due to the Brewer–Dobson circulation. The ozone cycle (Figure 3d) with highs in the austral summer (January-April) and lows in the austral winter (July-October) reflects the higher production of ozone in summer due to the peak of solar radiation compared to winter (Abdoulwahab, 2016). Apart from this natural cycle of stratospheric ozone at subtropical latitudes, no other spatio-temporal variation, potentially caused by Hunga volcano eruption, is visible at naked eye in Figure 3d. Some authors mention that, following Hunga volcano eruption, ozone concentrations increase in the middle

stratosphere and decrease in the lower stratosphere were caused by enhanced sulfate aerosol (Lu et al., 2023), others claimed that the midlatitude and tropics ozone reduction observed by MLS was mainly linked to circulation effects (Wang et al., 2022). Above the Indian Ocean Millet et al. (2024) reported an ozone mini-hole structure during the first week after the eruption. At this early stage of our understanding of the effects of Hunga volcano on the stratospheric ozone, the present study does not consider any potential increase/decrease of stratospheric ozone due to Hunga volcano eruption.

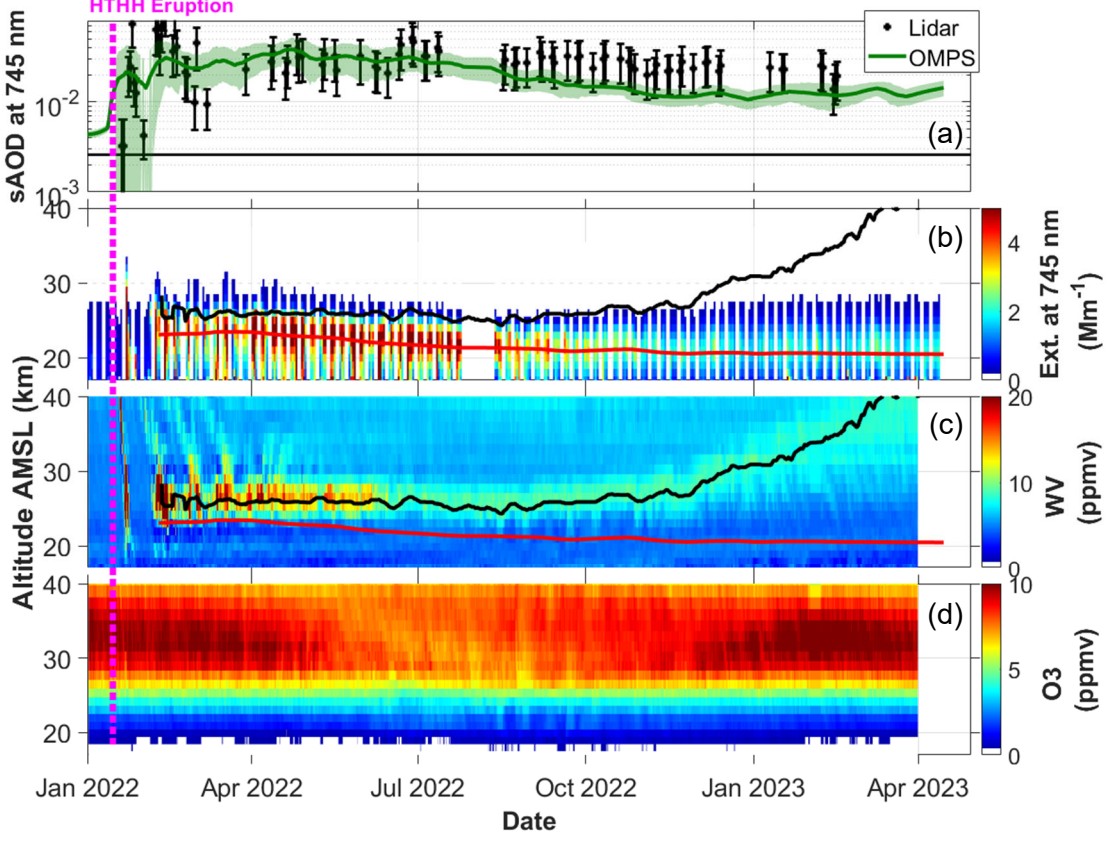

**Figure 3: (a) Monthly (OMPS) and instantaneous (lidar, nighttime) sAOD (17 – 40 km) at 745 nm over Reunion Island; Time-height plots of (b) extinction coefficient at 745 nm from OMPS, (c) water vapor and (d) ozone mixing ratio from M2-SCREAM. The green shaded area in (a) is the sum of the standard deviation (calculated as a 15-day rolling standard deviation) and the 10 % relative accuracy of OMPS at 745 nm according to Taha et al. (2021). The red and black lines in (b) and (c) report the peak height**
**of the aerosol and water vapor plumes, respectively.**

## 4 Impact of the Hunga volcanic plume on the Earth's radiation budget

In order to analyze the radiative impact of the aerosols and the water vapor, separately and altogether, 4 parametrizations of GAME are performed and summarized in Table 2. The perturbed condition is the full parametrization with observed sAOD and water vapor mixing ratio. For the unperturbed conditions, the impact of aerosols is assessed by assuming an aerosol-free

stratosphere; the impact of water vapor is assessed by assuming that the water vapor mixing ratio in the Hunga moist layer is

equal to the climatological value of 4.5 ppmv obtained from the MLS 2021 monthly means; the impact of aerosols and water vapor is assessed by assuming both an aerosol-free stratosphere and a water vapor mixing ratio of 4.5 ppmv in the Hunga moist layer.

Figure 4, Figure 5 and Figure 6 show the radiative impact of aerosols only, of the water vapor only, and of both aerosols and water vapor, respectively, in terms of time plots of $DRE(BOA)$ and $DRE(TOA)$ as well as time-height plots of heating/cooling (H/C) rate anomaly in the SW, LW and SW+LW spectral ranges. We analyze three different periods of time (see gray shaded areas in Figure 4), excluding from now on the first two weeks after the eruption to allow for some dispersion to happen:

- The entire period from Feb. 2022 to Feb. 2023 (M2 – M14, M1 being Jan. 2022).
- Feb. 2022 – Apr. 2022 (M2 – M4), the short-term period.
- May 2022 – Feb. 2023 (M5 – M14), the mid-term period.

The first period is representative of the radiative impact of Hunga volcano since the eruption to date, while the second and third periods are representative of the short- and mid-term tendencies, respectively. The radiative effects of the three simulations (aerosols only, WV only, aerosols and WV) associated to the three periods at the three atmospheric level (BOA, ATM, TOA) are summarized in Table 3. Figure 7 shows the averaged H/C rate anomaly profiles over both M2 – M4 and M5 – M14 periods and for the three simulations (aerosols only, WV only, aerosols and WV).

| Impact of… | Perturbed | Unperturbed |
|---|---|---|
| Aerosols | | sAOD = 0 <br> Measured WV |
| Water vapor | Measured sAOD <br> Measured WV | Measured sAOD <br> WV = 4.5 ppmv above 20 km (climatology from MLS monthly means in 2021) |
| Aerosol and water vapor | | sAOD = 0 <br> WV = 4.5 ppmv above 20 km (climatology from MLS monthly means in 2021) |

**Table 2. Aerosol and water vapor parametrizations for the perturbed/unperturbed simulations of GAME.**

*Aerosols* (Figure 4). In all moments the aerosol SW component dominates over the LW one at TOA (Figure 4a), producing a negative net radiative effect ($-1.10 \pm 0.45$ W m$^{-2}$). The net aerosol effect at TOA is stronger during M2 – M4 ($-1.37 \pm 0.58$ W m$^{-2}$) than during M5 – M14 ($-1.01 \pm 0.41$ W m$^{-2}$) which is caused by the sAOD decrease. Gupta et al. (2023) estimated a net instantaneous clear-sky radiative energy loss caused by an enhanced sAOD of $-0.48 \pm 0.04$ W m$^{-2}$ at TOA in the southern hemisphere. At BOA the aerosol LW component is nearly zero, so that the aerosol net $DRE$ is that of the SW component: a cooling is observed, stronger during M2 – M4 ($-1.26 \pm 0.50$ W m$^{-2}$) than during M5 – M14 ($-0.99 \pm 0.32$ W m$^{-2}$). As a

consequence, the atmosphere slightly cools: $DRE(ATM)$ = -0.11 ± 0.08 W m$^{-2}$ during M2 – M4 and -0.03 ± 0.09 W m$^{-2}$ during M5 – M14. It is extremely important to contrast those estimations to the uncertainties estimated in Section 2.3.2 (+0.09 / -0.06 W m$^{-2}$) as both the absolute magnitudes and their variability are in the same order of magnitude than this uncertainty. This reveals that the aerosol radiative effect has most likely contributed to cool the stratosphere during M2 –

M4, but the same conclusion is not valid for M5 – M14 because $DRE(ATM)$ with its uncertainty could equally be positive or negative. As a consequence of the neutral SW scattering and strong LW absorption, the time-height plots of H/C rate anomaly (Figure 4b-d) caused by the SW+LW aerosol impact is locally positive with daily heating rates (averaged over both the period considered and the altitude range 17 – 32 km, see Figure 7) of +0.05 ºK day$^{-1}$ (peaking at 26 km) during M2 – M4 and of +0.02 ºK day$^{-1}$ (peaking at 23 km) during M5 – M14. Our mid-term (M5 – M14) heating rate (+0.02 ºK day$^{-1}$) is in

quite good agreement with the annual-mean zonal value at 20º S averaged between 20 and 36 km, ∼ +0.02 ºK day$^{-1}$, estimated by Gupta et al. (2023). The time series of the H/C rate profiles (Figure 4d) follows the one of the aerosol extinction profiles with a decreasing tendency starting in April 2022 (Figure 3b).

*Water vapor* (Figure 5). The water vapor radiative effect is dominated by the cooling effect of water vapor longwave emission in the moist layer (Figure 5c), although the SW warming is not negligible in the first three months (Figure 5b). This

layer produces a slightly negative effect at TOA of -0.07 ± 0.02 W m$^{-2}$ during M2 – M4 which decreases to -0.03 ± 0.01 W m$^{-2}$ during M5 – M14. Sellitto et al. (2022) estimated also a negative $DRE(TOA)$ caused by water vapor for the fresh plume (instantaneous values of -0.7 and -0.4 W m$^{-2}$), but a positive $DRE(TOA)$ caused by water vapor for the aged plume (8 February 2022) of +0.8 W m$^{-2}$ and attributed to the descent in altitude of the moist layer. Our analysis supports neither this change of sign, nor this direction of the vertical motion of the moist layer over the long term. Zhu et al. (2022) who use a

global climate model to simulate the radiative effect in the first two months of 2022 find that when only water vapor is injected in their model (without sulfur dioxide; see their supplementary material) the zonal $DRE(TOA)$ at the latitude of Reunion Island is neutral-to-positive and much smaller than that caused by aerosols (their global mean over the first 2 months is -0.02 W m$^{-2}$). The discrepancy with our findings stems from an excess of water vapor in Zhu et al. (2022) simulations since the reaction of sulfur dioxide (not present) and hydroxide is not happening in their simulation; different

heights of the moist layer; and/or zonal vs. local computations. The water vapor radiative effect at BOA is negligible. The SW+LW water vapor radiative impact in the stratosphere is mostly negative with daily cooling rates (averaged over both the period considered and the altitude range 17 – 32 km, see Figure 7) of -0.07 ºK day$^{-1}$ (peaking at 26 km) during M2 – M4 and of -0.03 ºK day$^{-1}$ (peaking at 26 km) during M5 – M14. The same ascending behavior of the water vapor concentration (Figure 3c) is observed on the profiles of the SW+LW water vapor cooling rate (Figure 5d). Our mid-term (M5 – M14)

cooling rate (-0.03 ºK day$^{-1}$) is in quite good agreement with the annual-mean zonal value at 20º S averaged between 20 and 36 km, ∼ -0.015 ºK day$^{-1}$, estimated by Gupta et al. (2023). Schoeberl et al. (2022), who estimated the LW zonal impact of water vapor at 15º S for the first 6 months of year 2022, show a cooling effect in the stratosphere with a stronger peak at ∼ -0.5 ºK day$^{-1}$ at the end of February and decreasing afterwards. Sellitto et al. (2022) calculated strong instantaneous cooling rates peaking between -4.0 and -10 ºK day$^{-1}$ during the first two weeks after the eruption.

*Aerosols and water vapor* (Figure 6). The overall Hunga aerosol and water vapor impact on the Earth's radiation budget is negative (cooling) for the first thirteen months after the eruption: $DRE(TOA)$ = -1.14 ± 0.46 W m$^{-2}$ and the aerosols (WV) are responsible for ~ 96 (4) % of this cooling. The breakdown in short- and mid-term tendencies shows a decrease of the net $DRE(TOA)$ from -1.44 ± 0.60 W m$^{-2}$ during M2 – M4 to -1.04 ± 0.40 W m$^{-2}$ during M5 – M14. Locally, Sellitto et al. (2022) found an instantaneous aerosol and water vapor $DRE(TOA)$ negative in the first two weeks (ranging from -12.5 to -20.1 W

m$^{-2}$) and positive (+0.2 W m$^{-2}$) for what they call the "aged plume" (8 February 2022). At the global scale, Zhu et al. (2022), mentioned earlier, find a $DRE(TOA)$ of -0.21 W m$^{-2}$ for the first two months of 2022. Gupta et al. (2023) estimated a net instantaneous clear-sky radiative energy loss of -0.48 ± 0.06 W m$^{-2}$ at TOA in the southern hemisphere, resulting from its effects on stratospheric water vapor, aerosols, and ozone, and showed a certain zonal homogeneity over a one-year period. Our results regionalized to the southern tropical Indian Ocean are quantitatively in very good agreement with those of Gupta

et al. (2023). At the surface, a marked cooling produced by Hunga volcanic aerosols is observed ($DRE(BOA)$ = -1.05 ± 0.36 W m$^{-2}$ for the entire period), with a decreasing tendency with time ($DRE(BOA)$ = -1.26 ± 0.50 W m$^{-2}$ for M2 – M4 and -0.99 ± 0.32 W m$^{-2}$ for M5 – M14). At the global scale, Zhu et al. (2022) find a $DRE(BOA)$ of -0.21 W m$^{-2}$ for the first two months of 2022. Also Zuo et al. (2022) modelled the global surface temperature in the first year after the Hunga volcano eruption and found a negative anomaly of -0.004 ºK but recognized that it is "within the amplitude of internal variability at

the interannual time scale and thus not strong enough to have significant impacts on the global climate". By extension of our results to the southern tropical Indian Ocean region, our analysis shows that the eruption of Hunga volcano might have had a clear cooling impact on the regional climate at the surface in this region of the Earth. Finally, our results imply a slight loss of energy in the stratospheric volcanic layer which cools: the total atmospheric radiative budget is negative and reaches -0.18 ± 0.10 W m$^{-2}$ (60 % aerosols, 40 % water vapor) during M2 – M4 and -0.06 ± 0.09 W m$^{-2}$ (50 % aerosols, 50 % water vapor)

during M5 – M14. This cooling reflects apparently an overall similar effect of the aerosol and the water vapor on the stratosphere. However, it is against primordial to contrast those estimations to the uncertainties estimated in Section 2.3.2 (+0.09 / -0.06 W m$^{-2}$). Taking into account the absolute magnitudes and their uncertainty, one can claim that the aerosol and water vapor radiative effect has most likely contributed to cool the stratosphere since the eruption of Hunga volcano and that this effect might have fainted to negative-to-neutral by the end of the period considered. An interesting result at this point is

how the aerosols and water vapor H/C rates distribute vertically in the atmosphere. It is clear from Figure 6d that the negative longwave H/C rate caused by water vapor and the positive one caused by the aerosols coexist at different altitude levels. Like Figure 6d, Figure 7 shows that the aerosol and water vapor H/C rate anomaly (solid line) varies from positive (aerosols dominate) to negative (WV dominates) with increasing height. During M2 – M4, the H/C rate profile switches at ~22 km from positive (peak of +0.04 ºK day$^{-1}$ at 21 km) to negative (peak of -0.15 ºK day$^{-1}$ at 27 km) and presents a peak

ratio ~ 4. During M5 – M14, the H/C rate profile switches at ~24 km from positive (peak of +0.05 ºK day$^{-1}$ at 22 km) to negative (peak of -0.08 ºK day$^{-1}$ at 26 km) with a peak ratio ~ 1.5.

   Put in the literature context of other recent volcanic studies, the Hunga radiative effect at TOA is quite unique as far as the combined aerosol and WV impact is concerned, and the literature with which to compare our results is rather limited. A

general agreement is observed for the shortwave aerosol component which is systematically found negative in all studies
(volcano/year of the eruption): Raikoke/2019, Ulawun/2019 (Kloss et al., 2021), Ambae/2018 (Kloss et al., 2020), Kasatochi/2008, Sarychev/2009, Nabro/2011 (Andersson et al., 2015), and through multi-year (2000-2013) global volcanic aerosol forcing estimations (Ridley et al., 2014). For the longwave component, this is a completely different story and persuasive evidence of the LW volcanic forcing are far missing. For example, the method consisting of calculating the total as 70 % of the shortwave forcing like in Andersson et al. (2015) is definitely not recommended, at least not right after the
eruption. Schmidt et al. (2018) report that for eruptions after 2004, lower $SO_2$ emissions into lower altitudes (compared to Pinatubo) produce total shortwave radiative flux anomalies of comparable magnitude to the total LW forcing. Our work emphasizes the importance of including the aerosol and water vapor interaction with the longwave radiation in stratospheric volcanic studies, especially in the case of submarine eruptions. It also puts forwards the complexity to correctly estimate the TOA radiative forcing, and thus the ambiguity generated when different datasets, models, spatio-temporal averaging, etc. are

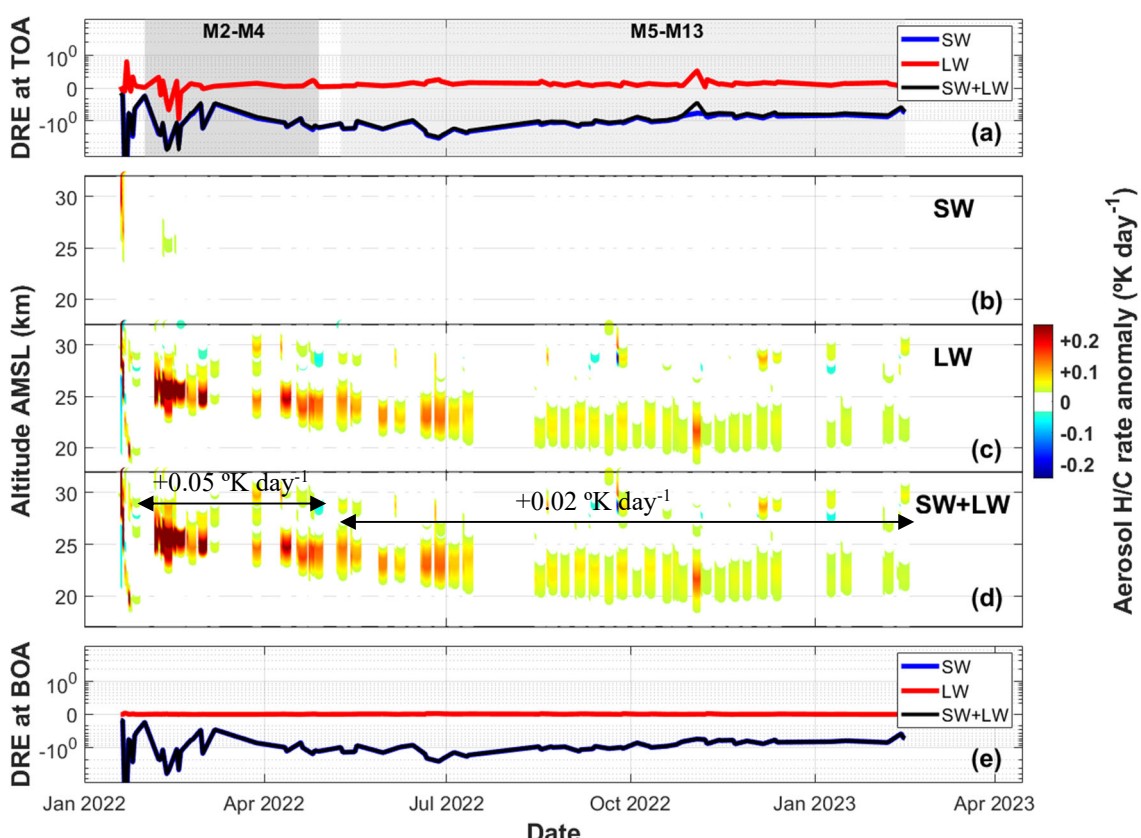

Figure 4: Aerosol direct radiative impact. (a) Radiative effect (W m$^{-2}$) at TOA; Time-height evolution of the H/C rate anomaly in the (b) SW; (c) LW; and (d) SW+LW spectral ranges; (e) Radiative effect (W m$^{-2}$) at BOA. In (a), the gray shaded areas represent the short- and mid-term periods: February to April 2022 (M2 – M4) and May 2022 to February 2023 (M5 – M14).

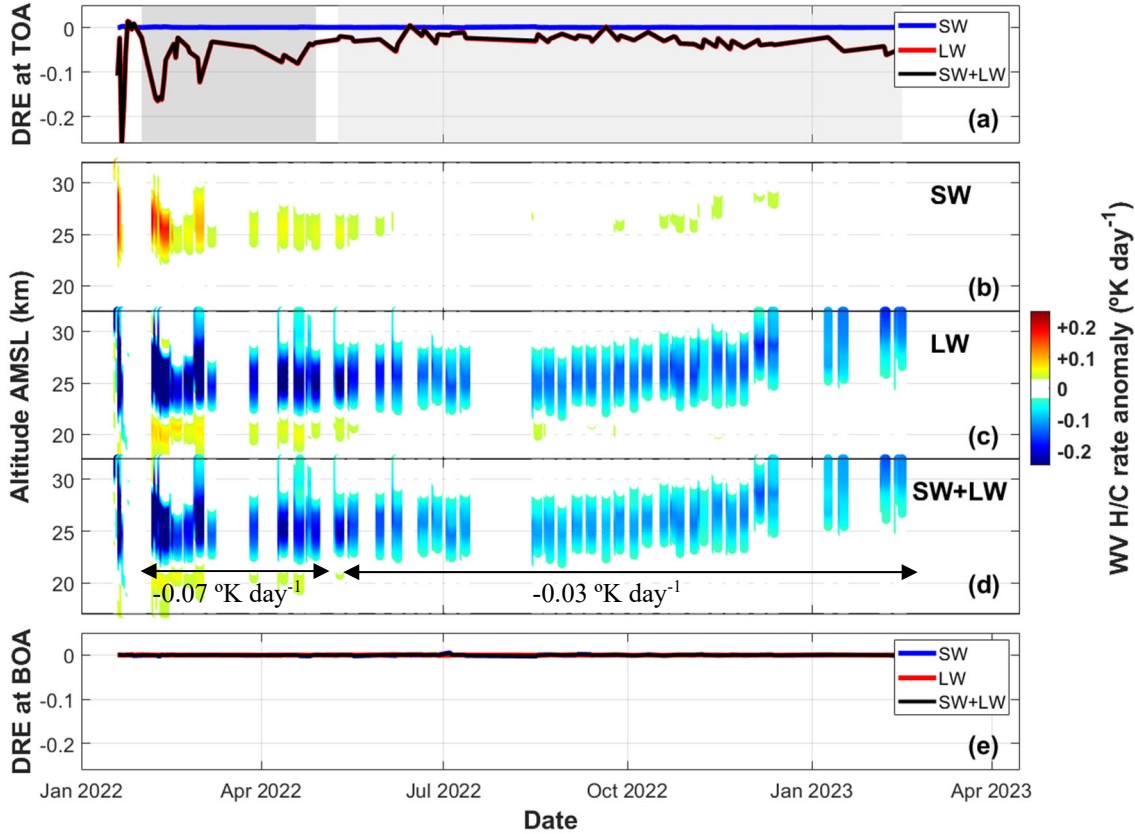

**Figure 5: Idem as Figure 4 for the water vapor. Note that the left axis scale in (a) and (e) is different from that of Figure 4.**

used. The use of unambiguous, direct measurements of radiative fluxes from space like in Minnis et al. (1993) are highly encouraged.


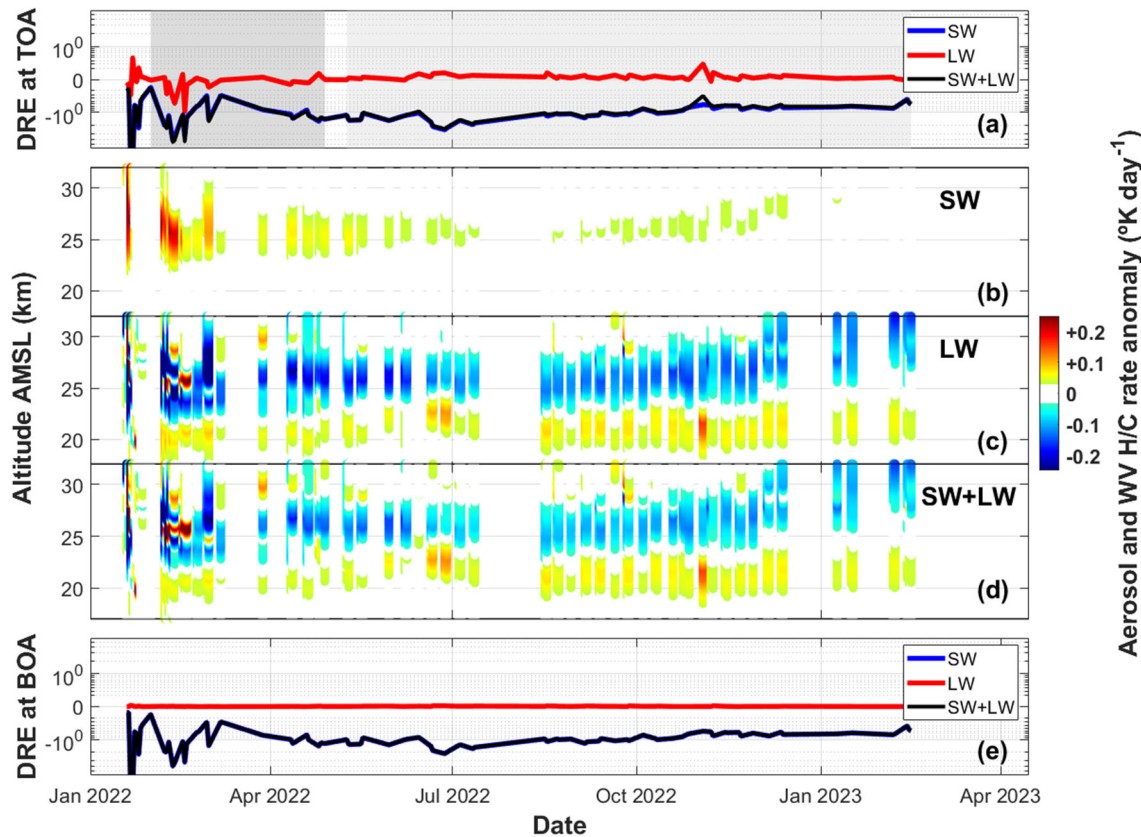

**Figure 6: Idem as Figure 4 for the aerosols and water vapor.**

|  | Aerosols | Water vapor | Aerosols and water vapor |
|---|---|---|---|
| Daily *DRE* (W m$^{-2}$) for the entire period M2 - M14 | | | |
| TOA | -1.10 ± 0.45 | -0.04 ± 0.02 | -1.14 ± 0.46 |
| ATM | -0.05 ± 0.10 | -0.04 ± 0.02 | -0.09 ± 0.10 |
| BOA | -1.05 ± 0.36 | < 0.01 | -1.05 ± 0.36 |
| Daily *DRE* (W m$^{-2}$) for the short-term period M2 – M4 | | | |
| TOA | -1.37 ± 0.58 | -0.07 ± 0.02 | -1.44 ± 0.60 |
| ATM | -0.11 ± 0.08 | -0.07 ± 0.02 | -0.18 ± 0.10 |
| BOA | -1.26 ± 0.50 | < 0.01 | -1.26 ± 0.50 |
| Daily *DRE* (W m$^{-2}$) for the mid-term period M5 – M14 | | | |
| TOA | -1.01 ± 0.41 | -0.03 ± 0.01 | -1.04 ± 0.40 |
| ATM | -0.03 ± 0.09 | -0.03 ± 0.01 | -0.06 ± 0.09 |
| BOA | -0.99 ± 0.32 | < 0.01 | -0.99 ± 0.32 |

**Table 3. SW+LW *DRE* at BOA, TOA and in the atmosphere produced by aerosols only, WV only and both aerosols and WV. These values are the average over the entire period M2 – M14, the short-term period M2 – M4, and the mid-term period M5 – M14, all excluding week 1 and 2 after the eruption.**

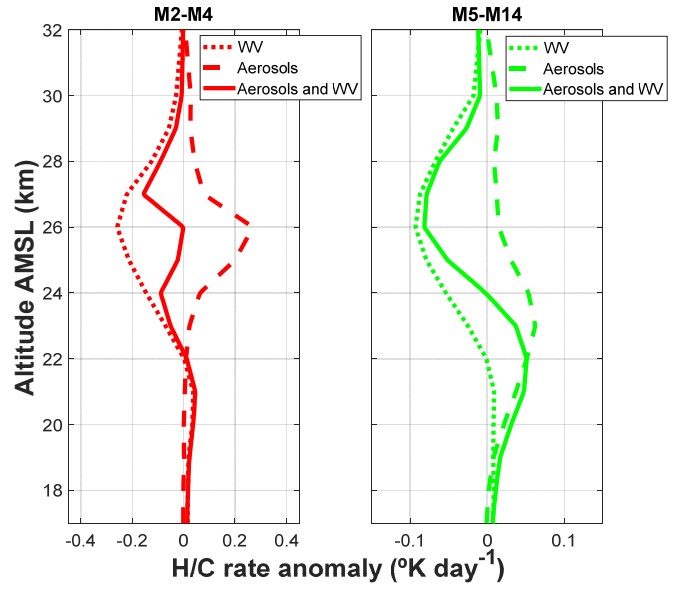

**Figure 7: Mean profiles of aerosols only, water vapor only, and aerosol and water vapor daily heating/cooling rate anomaly averaged over the short-term period (M2 – M4) and the mid-term period (M5 – M14), all excluding week 1 and 2 after the eruption.**

**5 Conclusions**

Thirteen months after the eruption of Hunga volcano, aerosols and water vapor are still present in the stratosphere of the southern tropical Indian Ocean region. During the first three months after the eruption the stratospheric aerosol optical depth increases and reaches a peak at 0.035 (~13 times the background sAOD) in April 2022, the highest in the last decade, and the third highest in the last 40 years (after Pinatubo and El Chichón). From April to November 2022 the sAOD decreases and then stabilizes at a value of 0.012 (~5 times the background sAOD). Unusually high water vapor concentrations are also observed in the stratosphere. On a monthly basis, the water vapor stratospheric peak reaches a maximum in February 2022 which is approximately a factor 5 above the climatological reference. In February 2023, this ratio has decreased down to almost 2.

In all moments, the water vapor plume is located at a higher altitude than the aerosol plume. The height of the peak of the aerosol and water vapor layers have opposite tendencies as of April 2022: the aerosol plume is slowly descending, mostly by gravitational settling, whereas the moist layer is ascending slowly until October 2022 and at a higher rate afterwards. The upward transport of the moist layer is due to the Brewer-Dobson circulation. Both aerosol and WV plumes are still present on 15 April 2023. The aerosol plume is located at 18.5 – 23.5 km height and the moist layer is above 30 km. As far as aerosols are concerned, the plume peak height decreases since April 2022 at an average steady rate of $\sim -0.008$ km day$^{-1}$. Assuming this rate constant in time, the remaining life time of the volcanic plume in the stratosphere is estimated to be between 2 and 2.5 years after 15 April 2023.

The radiative impact of both aerosol and water vapor layers is estimated at our site and assumed representative of the southern tropical Indian Ocean. Averages are made over 3 different periods of time in order to explain the temporal evolution: the first thirteen months, the short-term (M2 – M4) and the mid-term (M5 – M14) periods. During the first thirteen months after Hunga volcano eruption, the overall aerosol and water vapor impact on the Earth's radiation budget is negative (cooling, $-1.14 \pm 0.46$ W m$^{-2}$) and dominated by the aerosol impact (~96 %; the remaining ~4 % are due to WV). At the Earth's surface, aerosols are the main driver and produce a negative (cooling, $-1.05 \pm 0.36$ W m$^{-2}$) radiative impact. Between the short- and mid-term periods, the aerosol and water vapor radiative effect at both the surface and TOA reduces 21 and 28 %, respectively. Heating/cooling rate profiles show a clear vertical difference in the stratosphere during the mid-term period between the aerosol warming impact (17 to 25 km) and the water vapor cooling one (25 to 40 km). During the short-term period, a slight loss of energy of $-0.18 \pm 0.10$ W m$^{-2}$ is observed in the stratosphere with a balanced contribution between the aerosols (60 %) and the water vapor (40 %). During the mid-term period, this effect reduces to values in the same order of magnitude than the estimated uncertainty. This study shows that the eruption of Hunga volcano has had, so far, a clear radiative impact on the Earth's radiation budget in the southern tropical Indian Ocean region.

**Acknowledgments**

The authors from LACy acknowledge the support of the European Commission through the REALISTIC project (GA 101086690). This work was supported by CNES, through the projects EECLAT, AOS and EXTRA-SAT. The project OBS4CLIM (Equipex project funded by ANR: ANR-21-ESRE-0013) is acknowledged. The authors acknowledge the CNRS (INSU), Météo France, and Université de la Réunion for funding the infrastructure OPAR (Observatoire de Physique de l'Atmosphère à la Réunion) and OSU-R (Observatoires des Sciences de l'Univers à la Réunion, UAR 3365) for managing it. The federation Observatoire des Milieux Naturels et des Changements Globaux (OMNCG) of the OSU-R is also

acknowledged.

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
