# Peer review of "Radiative impact of the Hunga stratospheric volcanic plume: role of aerosols and water vapor in the southern tropical Indian Ocean"

_EGUsphere, 2024_

## Author Comment (AC1)

**RC1: 'Comment on egusphere-2024-1688'**

This paper calculates the radiative forcing of aerosol and water vapor volcanic cloud generated by the Hunga volcano eruption in January 2022. The calculations were performed for the specific location of Reunion Island. The perturbations of stratospheric aerosol were calculated, neglecting the effect of background aerosols, which can cause a 20-25% error; for water vapor, it was assumed that the unperturbed value is 4.5 ppmV for all altitudes, which was not precisely correct. The authors used a Line-by-Line radiative transfer model with the highest resolution of 20 cm-1 for radiative transfer calculations. That might be the course for resolving the effects of stratospheric water vapor, but it worked well. The authors extrapolated the imaginary part of the sulfate aerosol refractive index from near IR to visible and UV. As a result, they overestimated aerosol short wave (SW) absorption. This is especially well seen in stratospheric radiative heating, as the paper reports warming of the stratosphere during, e.g., the first four months after the eruption, while observations show significant cooling. The radiative forcing at the top of the atmosphere is reasonably correct, but SW aerosol radiative forcing at the bottom of the atmosphere (BOA) is exaggerated. These drawbacks have to be rectified before the paper can be published.

**Reply**: Thank you very much. We greatly appreciate the reviewer feedback and critical comments. In this new manuscript new estimations of the aerosol and WV radiative effects are presented. The greatest modification (wrt the initial estimations) is that the single scattering albedo in visible and UV has now been forced to 1 → no absorption, only scattering. This has had the effect of increasing (in absolute value) significantly the TOA radiative effect while reducing the BOA radiative effect, resulting now in a negative atmospheric (TOA-BOA) SW+LW radiative effect caused by the aerosols and the water vapor on the stratosphere. The discussion in Section 4 has also been changed accordingly.

As volcanologist recently updated the name of the volcano to "Hunga", the name was updated everywhere in the manuscript, including in the title.

Specific comments:

L38: The mass of water retained in the stratosphere was unprecedented, not the amount of emitted water.

**Reply**: Corrected.

L42: Do you mean at the location of Reunion Island or globally? I do not think it is right globally.

**Reply**: Some precisions have been brought in this sentence which now reads:

"Still, the stratospheric aerosol optical depth (sAOD) has been recorded **globally** as the largest since Pinatubo eruption (Taha et al., 2022) and peaked **locally** at values never observed before**, e.g. in the Indian Ocean** (Baron et al., 2023)."

L77: In this context, the reference should be "Jenkins et al. (2023)." Please correct the text in many other similar cases.

Reply: All references have been revised and the format Lastname et al. (yyyy) has been applied everywhere it was needed.

L108: Legrand et al. (2022) reported that the aerosol spatial distribution was patchy due to dynamic instabilities for more than six months.

Reply: The referee probably refers to Legras et al. (2022), as we have not found any article of Legrand et al. from 2022 about the HTHH. Legras et al. (2022) say that "volcanic sulfates and water still persisted after six months". It is true and our work shows that 14 months after the eruption volcanic sulfates and water still persist. Legras et al. (2022) also say that the aerosol spatial distribution was patchy due to dynamic instabilities, but only during the first 2 months (see their Section 6).

L190: Extrapolating the imaginary refractive index could cause spurious absorption in the UV and visible wave bands. It is well known that sulfate aerosols do not absorb in those wave bands.

Reply: In the dataset used, the last value of IRI at 2.36 μm is 4 x 10$^{-6}$ and it is this value (and not strictly 0) that has been assumed for IRI from 0.2 to 2.36 μm. Although very small, we realize now thanks to this comment that the associated SSA below 2.36 μm is indeed different from 1. We have now forced SSA in all shortwave spectral bands of GAME to the value of 1.00. The results are quite different. Fig. 4, 6 and 7 are new and the discussion has been totally revised.

Figure 1: Please show your aerosol LW SAOD for 10 um.

Reply: Figure 1 shows the spectral refractive index used for the calculation of the aerosol radiative properties. So, we are sorry to say that we don't understand this request at this place.

L236: You should use the word dispersion instead of dilution. SAOD is also defined by the rate of SO2 to SO4 conversion. OMPS-LP misses the initial stage of the SAOD generation, so it is not surprising that you see a discrepancy with OMPS observations at the initial stage.

Reply: Dispersion is used now everywhere in the manuscript instead of dilution.

L257: "zonal scale" - please clarify the sentence.

Reply: The sentence has been reformulated as:

"Such a difference, although not so accentuated, is observed zonally at 15º S during the first six months of year 2022 (Schoeberl et al., 2022)."

L263: Please be more specific.

**Reply**: The complement "although other mechanisms of volcanic aerosol removal exist" has been removed. Although it is true in a general sense, no other mechanisms in the case of HTHH are mentioned in Schoerberl et al. (2022).

L307: "probably correlated" > "caused"

**Reply**: Corrected.

L309: These results from (Zhu et al., 2022) cannot be used for comparison with your calculations, as without water vapor, volcanic clouds have different evolution and cannot be correctly interpreted.

**Reply**: The comparison with Zhu et al. (2022) for the aerosols has been removed.

Figure 7 shows Hunga's aerosol heating rate reaching 0.8 k/day in the first four months after the eruption, while after Pinatubo eruption the aerosol stratospheric heating rates were below 0.3 K/day. This cannot be right.

**Reply**: The new estimations of the radiative effect after the modifications mentioned in our first reply now show an aerosol stratospheric heating rate below 0.25 K/day for the period M2-M4 and below 0.07 K/day for the period M5-M14. Fig. 7 and the discussion have been changed accordingly.

L433-436: This conclusion about stratospheric warming contradicts observations that reported significant stratospheric cooling.

**Reply**: The new estimations of the radiative effect after the modifications mentioned in our first reply now show a slight stratospheric cooling.

---

## Author Comment (AC2)

**RC2: 'Comment on egusphere-2024-1688'**

This paper focusses on the characterization of stratospheric aerosols and water vapor over Reunion Islands in the southern tropical Indian Ocean. The manuscript associated variations in these two atmospheric parameters with the eruption of the Hunga Tonga-Hunga Ha' apai volcano on January 2022. The methodology used are based on backscattering lidar measurements emitting at 355 nm and on the Ozone Mapper and Profiler Suite Limb satellite. The Microwave Limb Sounder is also used for monthly mean water vapor. In general, the use of these instruments can serve sattisfactory for the purpuses of the study. Authors also claim the use MERRA-2 but it is not clear in the manuscript why they use this data. In generall, the authors presents very interesting measurements that are suitable for publications in Atmospheric Chemistry and Physics due to the possible impacts on the climate.

The manuscript also uses the measurements over Reunion Island in the GAME radiative transfer model for computing direct radiative transfer. I agree with the comments made by the previous referee. For aerosols, the input parameters have large uncertainties. For example, with the limited number of lidar measurements it is not possible the retrieval of aerosol microphysical properties that ultimately may affect GAME computations. I understand the proxies made by the authors, but it must be translated in error bars. This is a weak point that must be addressed before the final publication of the manuscript.

Reply: Thank you very much. We greatly appreciate the reviewer feedback and critical comments. An error budget has been calculated using propagation of errors of the lidar ratio and the radius of the size distribution considered on the aerosol radiative effect. A new Section "Error budget" has been added in the revised manuscript. See 7 comments ahead for a more detailed answer.

As volcanologist recently updated the name of the volcano to "Hunga", the name was updated everywhere in the manuscript, including in the title.

The authors in the last line of the conclusions claim that from this volcano eruption there is a clear impact on the regional climate of the Earh-Atmosphere system in the southern tropical Indian Ocean region'. To me this can not be deduced from the measurements and analyses performed in the manuscript. The tittle is confused as it suggests this impact on climate. I think that the title is incorrect and should be modified to reflect the purpursoes of the manuscript.

Reply: We have removed all conclusions on a possible "regional climate impact". The last sentence of the conclusion now reads:

"This study shows that the eruption of HTHH has had, so far, a clear radiative impact on the Earth's radiation budget in the southern tropical Indian Ocean region."

About the title of our paper "Radiative impact of the Hunga Tonga-Hunga Ha'apai stratospheric volcanic plume: role of aerosols and water vapor in the southern tropical Indian Ocean", we think it is quite appropriate with the purpose of the paper. No impact on climate is mentioned. If what is found confusing is the generalization of our results to the whole "southern tropical Indian Ocean region" (which is indeed justified at the end of Section 2.1), we ask the referee to say so, so that we can modify/remove this part of the paper, and then adapt the title of the paper accordingly.

Generally the manuscript is well-written, although there are many naive mistakes that must be improved to make the manuscript better legible:

Introduction Section: In general is very well, but I miss many references. For example:

Lines 33-34: Reference needed after "Several figures are evidences of a record-breaking atmosphere event". What are you referreing by 'Figures'

**Reply**: We mean "features". It has been corrected. These features are listed next with their corresponding references.

Lines 59-60: Reference needed after "Because ozone is not emitted primarly during volcanic eruptions, its loss or production by post-eruption reactions are more tedious to estimate"

**Reply**: The reference of Evan et al. (2023) has been added.

Lines 70-71: Reference needed after "… as volcanic sulfates are concerned, these aerosols usually scatter sunlight back to space, cooling the Earth´s atmosphere, and absorb outgoing thermal radiation"

**Reply**: The reference of Robock (2000) has been added.

Materials and Methods:

A brief overview is needed for this lidar – e.g. number of wavelengths, laser power, type of detection.

**Reply**: A full description of the lidar systems and their aerosol products at OPAR (Observatoire de Physique de l'Atmosphère à La Réunion) has just been accepted for publication in ESSD journal (Gantois et al., 2024). The last sentence of the first paragraph of Section 2.1 has been replaced by:

"A full description of the system is available in the data paper of Gantois et al. (2024)."

Gantois, D., Payen, G., Sicard, M., Duflot, V., Bègue, N., Portafaix, T., Marquestaut, N., Godin-Beekmann, S., Hernandez, P., and Golubic, E.: Multiwavelength, aerosol lidars at Maïdo supersite, Reunion Island, France: instruments description, data processing chain and quality assessment, Earth Syst. Sci. Data Discuss. [preprint], https://doi.org/10.5194/essd-2024-93, Accepted, 2024.

Authors mention 87 nights of measurements. How frequently are acquired the measurements.

**Reply**: The measurements are made twice a week on Monday and Tuesday nights. This information has been added in the first paragraph of Section 2.1.

Authors use 30 sr as lidar ratio. This is a potential source of errors because i) lidar ratio affect for the computation of the entire profile and ii) it might not be the real values. How do you accounts this possible source of uncertainties in GAME computations ?

Ozone Mapper and Profiler Suite Limb: Authors just use public data (that must be correctly referenced). But they are introducing additional errors in sAOD by forcing lidar to 745 nm for comparisons. Why not using 510 nm that is the closest wavelength to lidar measurements. If I am right, authors use $AE_{355/745}$ of -0.14 that might not be the real value for each specific measurement. That could add errors in direct radiative forcing computations.

The GAME radiative transfer model

I see that size distribution, single scattering albedo and assimetry parameters must be inputs and assumptions are made. This ok. But what is the final error in the computations? This could be computed assuming other aerosol optical and microphysical properties in the literature. Have you made these computations ?

**Reply**: We are answering in one place to the 3 comments above. We performed a sensitivity study on the lidar ratio and its associated error, and on the geometric median radius for which we assume a possible small decrease as observed by Duchamp et al. (2023). The results are compared to the nominal estimation and uncertainties in relative terms for TOA and BOA and in absolute terms for ATM are given in a new Table 1. This information fills in a new Section 2.3.2. Error budget. The sensitivity study on the geometric median radius results obviously in a change of all optical and radiative properties, including $AE_{355/745}$ . This new Section 2.3.2. reveals an uncertainty of $DRE(ATM)$ in the same order of magnitude than some of the $DRE(ATM)$ retrieved in the Section 4, and the results are discussed more cautiously in this respect. We copy paste here the new Section 2.3.2.

**2.3.2 Error budget**

An error budget is performed to quantify the uncertainties made on the radiative effect estimations using GAME and caused by the model itself, our parametrization and the hypothesis made. GAME model participated to an intercomparison exercise (Halthore et al., 2005) which concluded that it is accurate to a few units of watt (< 5 W) for a flux reaching 1000 W m$^{-2}$. The impact of this uncertainty on our estimations should be even less since only daily averaged fluxes are considered. It is thus reasonable to consider an uncertainty in relative terms of 0.5 %.

Two other sources of error are considered: one associated to the lidar ratio selected and another associated to the size distribution selected. The constant lidar ratio used in the elastic, 2-component inversion algorithm is 30 sr. Baron et al. (2023) estimated an uncertainty of ± 10 sr for the HTHH plume over Reunion Island in January 2022 (see Section 2.1). New profiles of the extinction inverted using (30 + 10) sr and (30 - 10) sr were used in GAME to quantify the deviation from the nominal (LR = 30 sr) radiative effect estimations. As far as the size distribution is concerned, Duchamp et al. (2023) detected "a small decreasing trend in the size" without quantifying it. We have assumed a decrease of the geometric median radius of -0.01 μm. Thus, a new Mie calculation was performed with a geometric median radius of 0.34 μm and the resulting radiative properties were used in GAME to quantify the deviation from the nominal (geometric median radius of 0.35 μm) radiative effect estimations. The results from these uncertainties are given in Table 1 in relative terms at BOA and TOA and in absolute

terms in the atmosphere. Logically, the lidar ratio error which impacts almost proportionally the sAOD error is by far the strongest. We can reasonably consider that the aerosol daily radiative effects are estimated with an uncertainty better than 48 % at TOA and better than 42 % at BOA. The resulting atmospheric radiative effect (TOA – BOA, see Eq. 2) is given with an uncertainty of +0.09 / -0.06 W m$^{-2}$.

| Source of error | TOA | BOA | ATM |
|---|---|---|---|
| GAME model | < + 0.5 % | < + 0.5 % | < + 0.5 % |
| LR (+10 / -10 sr) | +47 / -40 % | +42 / -38 % | +0.09 / -0.06 W m$^{-2}$ |
| Geometric median radius (-0.01 µm) | +4 % | ~0 % | < 0.01 W m$^{-2}$ |
| Total | +48 / -40 % | +42 / -38 % | +0.09 / -0.06 W m$^{-2}$ |

**Table 1. Error budget of the aerosol daily radiative effect.**

Results

Generally, I would like to point out a naive mistake: Many Figures are not introduced in the text and they just show up in the discussions. For a mature paper, every Figure must be appropiately introduced. The same happens for Tables. For example, in 282 says ' 4 runs of GAME are performed and summarized in Table', and when going to the Table I only find the configurations used in GAME.

**Reply**: We have been to all first calls of the figures and tables of the paper.

Fig. 1 (line 197) is properly introduced.

Fig. 2 (line 241), now introduced in the text.

Fig. 3 (line 259), now introduced in the text.

Fig. 4, 5 and 6 (line 314) are properly introduced.

Fig. 7 (line 325) is also properly introduced.

Table 1 (NEW, line 235) is properly introduced.

Table 2 (line 308) is properly introduced. We have changed the word "run" by "parametrization".

Table 3 (line 325) is already properly introduced.

Line 217: Background sAOD of 0.00259. How this value is computed ? I guess that the error associated with the measurements is larger than your standard deviations and might not have sense to give three significative values.

**Reply**: The background sAOD of 0.00259 is the mean of the monthly sAOD of the unperturbed years 20212 and 2013 (see Line 214 of the original manuscript). It is true that the error associated with the measurements is larger than the standard deviations found for this background sAOD. We have removed 1 digit and the background sAOD is now given as $(2.6 \pm 0.1) \times 10^{-3}$.

Line 237: The volcano also injected particles in the troposphere.

**Reply**: True, but tropospheric effects are out of the scope of this work.

Line 437-438: The study does not show the impact of HTHH on the regional climate in the southern tropical Indian Ocean region. To me, it deals with the aerosol and water vapor characterization plus radiative forcing computations. It might the impact claimed by the authors, but it can not be deduced from the results and discussions presented.

**Reply**: We have removed all conclusions on a possible "regional climate impact". The last sentence of the conclusion now reads:

"This study shows that the eruption of HTHH has had, so far, a clear radiative impact on the Earth's radiation budget in the southern tropical Indian Ocean region."

And I would like to add that I agree with the comments made by the other referee

**Reply**: All comments of RC1 have been taken into account in the revised manuscript. Please see the answers to that referee's comments.

---

## Author Comment (AC3)

**RC3: 'Comment on egusphere-2024-1688'**

The paper presents a good radiative characterization of the Hunga Tonga Hunga Ha'apai eruption. The authors present measurements and observations obtained at Reunion Island with Lidar and satellite measurements.

in the work an analysis of the results is presented in an analytical but very clear way making the paper clear and sequential in reading. Regarding the methodological part I think that some more details without having to resort to the references indicated would have been useful to make the reader easily informed on the observational capabilities. Specifically a more exhaustive description of the lidar system would give the reader the possibility to understand the characteristics and observational potential of the Reunion observatory.  Even a few brief additions on why certain assumptions were chosen in the data analysis would have provided the reader who is not an expert in Lidar with a more comprehensive explanation of the work (e.g. Line 91 LR=30).

**Reply**: Thank you very much. We greatly appreciate the reviewer feedback and critical comments.

A more exhaustive description of the lidar system has also been requested by Referee #2. A full description of the lidar systems and their aerosol products at OPAR (Observatoire de Physique de l'Atmosphère à La Réunion) has just been accepted for publication in ESSD journal (Gantois et al., 2024). The last sentence of the first paragraph of Section 2.1 has been replaced by:

"A full description of the system is available in the data paper of Gantois et al. (2024)."

Gantois, D., Payen, G., Sicard, M., Duflot, V., Bègue, N., Portafaix, T., Marquestaut, N., Godin-Beekmann, S., Hernandez, P., and Golubic, E.: Multiwavelength, aerosol lidars at Maïdo supersite, Reunion Island, France: instruments description, data processing chain and quality assessment, Earth Syst. Sci. Data Discuss. [preprint], https://doi.org/10.5194/essd-2024-93, Accepted, 2024.

In addition, a sentence summarizing the finding of Baron et al. (2023) for justifying the choice of the lidar ratio at 355 nm has been added. It reads:

"Indeed the latter found values of LR at 355 nm in the range 29 – 35 sr with small standard deviations (< 7 sr) by applying the transmittance method during several nights in January 2022."

Finally, as volcanologist recently updated the name of the volcano to "Hunga", the name was updated everywhere in the manuscript, including in the title.

From my point of view therefore the work is important to be published also given the low frequency of these events which as illustrated by the authors see in literature still relevant presentations of the eruption of Pinatubo and El Chichon underlining to the scientific community the importance of these ground and satellite observation systems for the study and characterization of these events.

**Reply**: We do also hope that our work will be published in order for our results to serve as constraint reference points for future works estimating HTHH forcing impact at larger scales.

---

## Author Response (AR2)

**RC1: 'Comment on egusphere-2024-1688'**

The paper calculates the radiative forcing at Reunion Island in the southern tropical Indian Ocean. The radiative transfer calculations use lidar observations combined with reanalysis data. The methodology sounds. The results could be helpful for a deeper understanding of physical processes and validating the model output. In the revised version, the results are reasonable except for the aerosol LW DRE at TOA (Fig. 4a), Which cannot be negative. This should be corrected. However, the paper is poorly written and suffers from multiple misinterpretations. The radiative forcing over Reunion Island is correlated with the hemispheric radiative forcing but is not representative of the entire Southern Indian Ocean, as the authors suggest. The volcanic cloud homogenized quickly in the zonal direction, so it would be correct to talk about the forcing in the Southern Hemisphere. The poleward dispersion in the stratosphere is slow, and significant latitudinal gradients of stratospheric aerosols and water vapor remain longer. Consequently, the radiative forcing calculated in this study is 2-3 times bigger than the hemispheric mean forcing calculated in other studies.

**Reply**: The referee spotted a hidden shortcoming of our analysis. Thank you very much for the detailed review. Yes, indeed LW DRE(TOA) cannot be negative. Several tries were made in which we wanted to test the implication of subtracting a non-zero aerosol background in the stratosphere. Those tests gave mostly positive LW DRE(TOA) with occasional negative values. And one of those tests was used in the original manuscript. We are deeply sorry for the inconvenience. When forcing sAOD to zero in the unperturbed simulations, as it has been stated from the very beginning, we obtain strictly positive LW DRE(TOA). This has also the effect of reducing the SW+LW DRE(TOA) which now agrees better with hemispheric studies (see Gupta et al. (2023) and a comment of the referee ahead). All calculations for the Aerosols and for the Aerosols and water vapor cases have been redone. Figures 4, 6 and 7 as well as Table 2 have been updated accordingly.

We approve the referee's statement to say that our site is not representative of the entire Southern Indian Ocean. The last paragraph of Section 2.1 has been removed. We now state that the in the revised manuscript that our results as representative of Reunion Island or at the most of the zonal hemisphere at 21º S.

With the overall review of the manuscript, the new calculations in the LW for the aerosols and the elimination of the discussion about DRE(ATM) which indeed made no sense without including the troposphere (see a couple of comments of the referee on that topic), we now hope that most of the misinterpretations have disappeared and that the manuscript has reached sufficient quality to be published in ACP.

Specific comments:

L23: Water vapor practically does not have any radiative effect at the surface.
**Reply**: The sentence "Water vapor has hardly any radiative effect at the surface." has been added in the abstract.

L24-26: Short-term and long-term periods are not defined.
Reply: These periods are defined in terms of months after the eruption. We have also added the periods now.

L33: Volcanic clouds cannot be dispersed from pole to pole in three months.
Reply: We have re-read Taha et al., 2022 and have smoothen this sentence which now reads: "The event showed a fast spatio-temporal, global dispersion of the stratospheric volcanic matter that circulated the Earth in only one week (Khaykin et al., 2022) **with small parts of the main aerosol layer** dispersed pole-to-pole in three months (Taha et al., 2022), first in the form of concentrated patches (Legras et al., 2022).".

L40: 37 Tg was probably retained in the stratosphere as a result of cross-tropopause transport. Evidence shows that Pinatubo injected more than 100 Tg of water vapor, which was quickly sedimented.
Reply: The sentence has been rewritten as: "The mass of water retained in the atmosphere was unprecedented: (Millán et al., 2022) estimated to 146 Tg the mass of water injected in the stratosphere (.e.g. 37 Tg of water was retained in the stratosphere as a result of cross-tropopause transport after the 1991 Pinatubo eruption (Pitari and Mancini, 2002)).".

L44-45: It is misleading to make a comparison in one point. Of course, Pinatubo and El Chichon produced higher concentrations of SO2 and SO4 locally.
Reply: The last part of this sentence has been deleted.

L50: Larger in comparison with what?
Reply: Larger compared to low-concentration volcanic sulfate that do not coagulate. The sentence has been modified as "Higher concentrations of volcanic sulfate led to more rapid coagulation and thus to particles quickly growing in size.".

L51: Faster SO2-to-SO4 conversion causes faster growth (in time) of sAOD but cannot change the total sAOD if the same mass of SO2 is converted.
Reply: The change in sAOD is caused by the coagulation mechanism which is enhanced in dense plumes, as evidenced by Zhu et al., 2022.

L63-74: Please be more specific. Water vapor emits and absorbs LW radiation. The injection height is an important factor, but not a driver. Sulfate aerosols absorb LW and SW near IR. Schoeberl et al. (2024) extended the analysis to 2 years.
Reply: This paragraph has been replaced by "In particular, the climate forcing will depend on the radiative effect produced by the water vapor longwave emission and absorption, and the sulfate aerosol longwave and shortwave near-infrared absorption (Robock, 2000). These interaction mechanisms

(emission and absorption) with the longwave and shortwave near-infrared radiation are height-dependent and determine the sign of the differential of energy gained (positive) or lost (negative) in all layers of the atmosphere.". The reference of Schoeberl et al., 2024 is very interesting. Thank you. It has been added also.

L75: More uncertain than what? Greater sensitivity to variations of what?
**Reply**: This sentence has been removed.

L92: remove "coefficient."
**Reply**: Corrected.

L101: What uncommon properties? It was sulfate aerosols most of the time.
**Reply**: The uncommon adjective refers to the lidar ratio values found in the range 29-35 sr, which are not common values for sulfate aerosols. This sentence was reformulated as "These uncommon lidar ratio values for sulfate aerosols were proved to be stable over time by Duchamp et al. (2023) using SAGE-III (Stratospheric Aerosol and Gas Experiment) observations.".

L113: It is not a scientific argument. Please remove.
L115-122: Why is this important if lidar observes the stratosphere? Does the retrieval use the AERONET observations?
**Reply**: We agree with the referee, in view also of the recent publication of Schoeberl et al. (2024), that a clear latitudinal gradient exist. We do not generalize anymore our results to the southern tropical Indian Ocean region. The paragraph mentioned has been deleted. The discussions about the regionalization of our results in the text and in the conclusions have been deleted. Consequently, the title has been modified as follows: "Radiative impact of the Hunga stratospheric volcanic plume: role of aerosols and water vapor over Reunion Island (21º S, 55º E)".

The second comment about L115-122 becomes therefore irrelevant.

L137: It should be "parameterize." It is written correctly in one place and incorrectly in many others.
**Reply**: Corrected.

L177: DISORT requires a phase function, not an asymmetry parameter. Do you use a parameterization for the phase function?
**Reply**: The DISORT module in GAME is parameterized with the Henyey-Greenstein analytic formula that approximates the shape of the actual phase function as a function of the asymmetry factor.

L198: What is the value of the imaginary refractive index in the 0.2-2.36 um range?
**Reply**: Its value has been forced to 1, so as to answer to Referee #1 and #2 of the initial review.

L220-221: "budget" > "estimate"
**Reply**: Corrected.

L226-235: Repetition
**Reply**: The second sentence of this paragraph has been deleted. The rest has not been said before.

L236: "strongest" > "largest"
**Reply**: Corrected.

L250-253: It is misleading to make this comparison in one point for volcanoes located in different places.
**Reply**: The comparison is made for estimations of global sAOD. It has been emphasized in the sentence which now reads: "Further back historically, the 40+ year satellite record of monthly sAOD at the scale of the globe (i.e. for the 60º S – 60º N latitude band) in Khaykin et al. (2022) shows that only the eruptions of Pinatubo (1991) and El Chichón (1982) exceeded the Hunga one in terms of absolute stratospheric AOD (by a factor of 6 and 3, respectively).".

L265-266: This proves homogenization in the zonal direction. We know it is fast.
**Reply**: "global dispersion" has been replaced by "dispersion in the zonal direction" in this part of the manuscript which now reads: "This time difference is an indication of the dispersion time of the volcanic matter injected by Hunga volcano in the stratosphere in the zonal direction. Other studies confirm that some parts of the volcanic plume dispersed pole-to-pole in three months (Khaykin et al., 2022; Taha et al., 2022). Another indicator of this dispersion in the zonal direction is the standard deviation (calculated as a 15-day rolling standard deviation) associated to OMPS monthly sAOD: once passed the first month, it steadily decreases all along year 2022.".

L268: No
**Reply**: The sentence has been deleted. See the answer to the comments about Line 113 and 115-122.

L273: Remove "coefficient."
**Reply**: Corrected.

L336:This is misleading. DRE(ATM) reflects absorption in the entire atmospheric column. But sulfate aerosols warm the stratosphere and cool the troposphere. You cannot use it to characterize the impact on the stratosphere. Look at your heating rates (Fig. 7).

L339: I'm afraid that's not right. The stratospheric sulfate aerosols warm the stratosphere (see Fig. 7).

**Reply**: We thank the referee for this important comment. Without the analysis in the troposphere, we have decided to remove all statements about DRE(ATM) which cannot be inferred from our current analysis. Indeed stratospheric sulfate aerosols warm the stratosphere and this is now shown only from the analysis of the H/C rate profiles (Figure 4 and Figure 7).

L349: And absorption

**Reply**: Corrected.

L368-369: These high heating rates are inconsistent with your calculations

**Reply**: This comparison with Sellitto's strong estimations in the plume just 2 weeks after the eruption have been deleted in the revised manuscript.

L379: Your forcing is 2-3 times bigger than in Gupta et al.

L385: You cannot bluntly extend your results to the entire region. I agree that radiative forcing at Reunion Island correlates with the hemispheric mean forcing. Still, qualitatively, it is 2-3 times larger, manifesting that the aerosol distribution is not latitudinally uniform.

**Reply**: By presenting in the revised manuscript our results as representative of Reunion Island or at the most of the zonal hemisphere at 21º S, the explanation of the differences observed with Gupta et al. (2023) is more evident. Also, the revision of our LW calculations lead now to a smaller difference between our results and Gupta's. This sentence has been modified as follows: "Our results, representative of Reunion Island and likely of the zonal hemisphere at 21º S, are a little less than twice larger than those of Gupta et al. (2023) manifesting that the aerosol distribution is not latitudinally uniform.".

L388: You cannot make conclusions about the "stratospheric volcanic layer" using DRE(ATM) because it characterizes the entire atmospheric column.

**Reply**: We thank the referee for this important comment. Without the analysis in the troposphere, we have decided to remove all statements about DRE(ATM) which cannot be inferred from our current analysis.

L395-400: What is this about? Please clarify.

**Reply**: With the new Figure 7, this part has been completely re-written. It now reads:

"It is clear from Figure 6d that the negative longwave H/C rate caused by water vapor and the positive one caused by the aerosols coexist at different altitude levels. During M2 – M4, the small height

difference between the sulfate and the moist layers and the higher rate of cooling of the latter result in an aerosol and water vapor H/C rate negative in most of the altitude range considered (red, solid line in Figure 7). During M5 – M14, H/C rate profiles show a clear vertical difference locally in the stratosphere between the aerosol warming impact (18 to 26 km) and the water vapor cooling (22 to 30 km). The resulting aerosol and water vapor H/C rate profile follows a S-shaped curve with peaks slightly larger for the moist layer (-0.09 ºK day$^{-1}$ at 26 km) than for the sulfate layer (+0.06 ºK day$^{-1}$ at 22 km.”

L407-414: This is misleading. The LW effect of stratospheric sulfate aerosols is well understood and calculated properly. Do not miss it with cases in the past when SW forcing was observed, but LW was not, and there is no information to reconstruct it. Then, people have to make assumptions.

**Reply**: We have changed this part of the paragraph to fit to the referee's suggestion which is actually reflected in the work of Schmidt et al. (2018) already cited in the manuscript. Now this part reads: "For the longwave component, persuasive evidence of the volcanic longwave effect has been missing for a long time in the past. However, the longwave effect of stratospheric sulfate aerosols is now well understood and calculated properly (Schmidt et al., 2018).".

---

## Author Response (AR3)

**Report #1 submitted on 16 Nov**

The revised paper has been significantly improved compared to its first versions, and I am happy to recommend it for publication. It is a valuable study that conducts radiative transfer calculations based on observations and clears up some inconsistencies from the prior publications regarding the radiative forcing at the Reunion Island site. I have a few minor remarks that might help to improve the manuscript further.

General comment:

What the authors call Direct Radiative Effect (DRE) is Instantaneous Radiative Forcing (IRF). It should not be mixed with an Adjusted Radiative Forcing (ARF). For example, Zhu et al. (2022) presented ARF and IRF. IRF is in the supplement. Please have this in mind when making comparisons with other studies.

Reply: Thank you for the advice. We just realized this difference of nomenclature, thank you. It is true that our DRE is to be compared with Zhu et al. IRF. Some slight changes have been made in this sense in the manuscript. It is also worth mentioning that the "instantaneous" adjective could result ambiguous in some circumstances since it could also refer to the temporal resolution of the DRE estimation. In our work, daily DRE are presented.

Specific comments:

L54: Are these diameters or radii?

Reply: radius. It has been added.

L114: I am not sure if OMPS was introduced.

Reply: It is now done Line 114 and Line 108 where OMPS appears for the first time.

L260: If you said that the descent is 244 m per month, keeping the same units and saying it is 8 m per day is better.

Reply: Done here, in the abstract and conclusions.

L323: It is an overstatement, as positive forcing is far within error bars.

Reply: on Line 323, the statement made is about the H/C rate anomaly, not the forcings.

L340: I am sure Zhu et al. (2022) calculate the sulfur cycle where OH oxidizes SO2.

Reply: This is true that Zhu et al. calculate the sulfur cycle. But the comparison here is made against Zhu et al. simulations called "H2Oonly – control" (supplementary Figure 10, bottom line, second column for the TOA Net Instantaneous). In this simulation of Zhu, no SO2 is considered.

L356: It is not essential, but here, DRE is compared with ARF but should be with IRF. Their IRF is 0.19 W/m2.

Reply: Changed.

L362: decreasing IN ABSOLUTE value ...

Reply: Done.

L426: It is overstatement. The aerosol lifetime is shorter because there are other removal processes.

Reply: The beginning of the sentence now reads "Assuming this rate constant in time and omitting other removal processes, the remaining life time…"